# Exact Gaussian Processes on a Million Data Points

Ke Alexander Wang[1][*]    Geoff Pleiss[1][*]    Jacob R. Gardner[2]
Stephen Tyree[3]    Kilian Q. Weinberger[1]    Andrew Gordon Wilson[1,4]
[1]Cornell University, [2]Uber AI Labs, [3]NVIDIA, [4]New York University

## Abstract

Gaussian processes (GPs) are flexible non-parametric models, with a capacity that grows with the available data. However, computational constraints with standard inference procedures have limited exact GPs to problems with fewer than about ten thousand training points, necessitating approximations for larger datasets. In this paper, we develop a scalable approach for exact GPs that leverages multi-GPU parallelization and methods like linear conjugate gradients, accessing the kernel matrix only through matrix multiplication. By partitioning and distributing kernel matrix multiplies, we demonstrate that an exact GP can be trained on over a million points, a task previously thought to be impossible with current computing hardware, in less than 2 hours. Moreover, our approach is generally applicable, without constraints to grid data or specific kernel classes. Enabled by this scalability, we perform the first-ever comparison of exact GPs against scalable GP approximations on datasets with $10^4 - 10^6$ data points, showing dramatic performance improvements.

## 1   Introduction

Gaussian processes (GPs) have seen great success in many machine learning settings, such as black-box optimization [38], reinforcement learning [6, 8], and time-series forecasting [33]. These models offer several advantages – principled uncertainty representations, model priors that require little expert intervention, and the ability to adapt to any dataset size [31, 32]. GPs are not only ideal for problems with few observations; they also have great promise to exploit the available information in increasingly large datasets, especially when combined with expressive kernels [41] or hierarchical structure [5, 35, 43, 45].

In practice, however, exact GP inference can be intractable for large datasets, as it naïvely requires $\mathcal{O}(n^3)$ computations and $\mathcal{O}(n^2)$ storage for $n$ training points [32]. Many approximate methods have been introduced to improve scalability, relying on mixture-of-experts models [7], inducing points [12, 37, 39, 42], random feature expansions [20, 30, 47], or stochastic variational optimization [2, 16, 17, 36, 46]. However, due to the historical intractability of training exact GPs on large datasets, it has been an open question how approximate methods compare to an exact approach when much more data is available.

In this paper, we develop a methodology to scale exact GP inference well beyond what has previously been achieved: we train a Gaussian process *on over a million data points*, performing predictions *without approximations*. Such a result would be intractable with standard implementations which rely on the Cholesky decomposition. The scalability we demonstrate is made feasible by the recent Blackbox Matrix-Matrix multiplication (BBMM) inference procedure of Gardner et al. [11], which uses conjugate gradients and related methods to reduce GP inference to iterations of matrix multiplication. Gardner et al. [11] show that this procedure (1) achieves exponential convergence using a pivoted Cholesky preconditioner under certain conditions, (2) requires relatively few number

---

of conjugate gradient steps to convergence for typical datasets, and (3) can more accurately solve linear systems than Cholesky-based approaches. Using BBMM, our approach is generally applicable without constraints to input grids or specific kernel families.

By partitioning and distributing kernel matrix multiplications across GPUs, we reduce the memory requirement for GP training to $\mathcal{O}(n)$ *on an individual GPU*, permitting scaling beyond $n \approx 10^4$ samples. Additionally, we introduce a number of practical heuristics to accelerate training and maximally utilize parallelization. With 8 GPUs, GPs can be trained in seconds for $n \approx 10^4$, hours for $n \approx 10^5$, and a few days for $n \approx 10^6$. In addition, we show that the training time can be further reduced by using better hyperparameter initializations. Nevertheless, all trained models can *make exact GP predictions in less than 1 second on 1 GPU* using a simple caching strategy.

We benchmark on regression datasets from the UCI repository [1]. We find exact GPs offer notably better performance on these datasets, often exceeding a two-fold reduction in root-mean-squared error. The results show how non-parametric representations continue to significantly benefit from the addition of new training points, a valuable conceptual finding in favor of non-parametric approaches. These results clarify the relative performance of popular GP approximations against exact GPs in an unexplored data size regime and enable future comparisons against other GP approximations. Further, they serve as a signpost to practitioners using GPs on big data and set the stage for theorists to propose better approximations to address this gap in performance.

## 2  Background

**Gaussian processes** (GPs) are non-parametric machine learning models that place a distribution over functions $f \sim \mathcal{GP}$. The function distribution is defined by a *prior mean function* $\mu : \mathbb{R}^d \to \mathbb{R}$, a *prior covariance function* or *kernel* $k : \mathbb{R}^d \times \mathbb{R}^d \to \mathbb{R}$, and observed (training) data $(X, \mathbf{y})$. The choice of $\mu$ and $k$ encode prior information about the data. $\mu$ is typically chosen to be a constant function. Popular kernels include the RBF kernel and the Matérn kernels [32].

*Notation:* Throughout this paper we will use the following notation: given training inputs $X \in \mathbb{R}^{n \times d}$, $K_{XX}$ is the $n \times n$ kernel matrix containing covariance terms for all pairs of entries. The vector $\mathbf{k}_{X\mathbf{x}^*}$ is a vector formed by evaluating the kernel between a test point $\mathbf{x}^*$ and all training points. $\widehat{K}_{XX}$ is a kernel matrix with added Gaussian observational noise (i.e. $\widehat{K}_{XX} = K_{XX} + \sigma^2 I$).

*Training:* Most kernels include hyperparameters $\theta$, such as the lengthscale, which must be fit to the training data. In regression, $\theta$ are typically learned by maximizing the GP's *log marginal likelihood* with gradient descent:

$$\mathcal{L} = \log p(\mathbf{y} \,|\, X, \theta) \propto -\mathbf{y}^\top \widehat{K}_{XX}^{-1} \mathbf{y} - \log |\widehat{K}_{XX}|, \tag{1}$$

$$\frac{\partial \mathcal{L}}{\partial \theta} \propto \mathbf{y}^\top \widehat{K}_{XX} \frac{\partial \widehat{K}_{XX}^{-1}}{\partial \theta} \widehat{K}_{XX} \mathbf{y} - \mathrm{tr}\left\{ \widehat{K}_{XX}^{-1} \frac{\partial \widehat{K}_{XX}}{\partial \theta} \right\}. \tag{2}$$

A typical GP has very few hyperparameters to optimize and therefore requires fewer iterations of training than most parametric models.

*Predictions:* For a test point $\mathbf{x}^*$, the GP predictive posterior distribution $p(f(\mathbf{x}^*) \mid X, \mathbf{y})$ with a Gaussian likelihood is Gaussian with moments:

$$\mathbb{E}\left[f(\mathbf{x}^*) | X, \mathbf{y}\right] = \mu(\mathbf{x}^*) + \mathbf{k}_{X\mathbf{x}^*}^\top \widehat{K}_{XX}^{-1} \mathbf{y} \tag{3}$$

$$\mathrm{Var}\left[f(\mathbf{x}^*) | X, \mathbf{y}\right] = k(\mathbf{x}^*, \mathbf{x}^*) - \mathbf{k}_{X\mathbf{x}^*}^\top \widehat{K}_{XX}^{-1} \mathbf{k}_{X\mathbf{x}^*} \tag{4}$$

Portions of these equations can be precomputed as part of training to reduce the test-time computation. In particular, (3) is reduced to an $\mathcal{O}(n)$ matrix-vector multiplication once $\widehat{K}_{XX}^{-1} \mathbf{y}$ is computed and cached. Similar caching techniques can reduce the asymptotic time complexity of (4) as well [28].

**The Cholesky decomposition** is used in many GP implementations to compute $\widehat{K}_{XX}^{-1} \mathbf{y}$, $\log |\widehat{K}_{XX}|$, and $\mathrm{tr}\left\{ \widehat{K}_{XX}^{-1} \left( \partial \widehat{K}_{XX} / \partial \theta \right) \right\}$ in (1) and (2). The positive definite kernel matrix $\widehat{K}_{XX}$ can be factorized into $LL^\top$, where $L$ is lower triangular. Computing $L$ requires $\mathcal{O}(n^3)$ time and $\mathcal{O}(n^2)$ memory. After computing this factorization, matrix solves and log determinants take $\mathcal{O}(n^2)$ and $\mathcal{O}(n)$ time respectively. The columns of $L = \begin{bmatrix} \mathbf{l}^{(1)} & \ldots & \mathbf{l}^{(k)} \end{bmatrix}$ are computed recursively [14]. Although concurrent work by [26] used the Cholesky decomposition for large scale GP inference through distributed

computing, it requires quadratic communication costs and quadratic memory. Furthermore, its recursive nature makes the Cholesky algorithm less amenable to GPU acceleration since GPUs are designed to parallelize matrix-vector multiplications.

**Conjugate gradients** (CG) is an alternative method for computing $\widehat{K}_{XX}^{-1}\mathbf{y}$. CG frames $\widehat{K}_{XX}^{-1}\mathbf{y}$ as the solution to an optimization problem: $\mathbf{v}^* = \arg\min_{\mathbf{v}} \frac{1}{2}\mathbf{v}^\top \widehat{K}_{XX}\mathbf{v} - \mathbf{v}^\top \mathbf{y}$, which is convex by the positive-definiteness of $\widehat{K}_{XX}$. The optimization is performed iteratively, with each step requiring a matrix-vector multiplication with $\widehat{K}_{XX}$. For a specified tolerance $\epsilon$ of the relative residual norm $\|\widehat{K}_{XX}\mathbf{v}^* - \mathbf{y}\|/\|\mathbf{y}\|$, the solution can be found in $t_\epsilon$ iterations. The exact number of iterations depends on the conditioning and eigenvalue distribution of $\widehat{K}_{XX}$, but $t_\epsilon \ll n$ for reasonable values of $\epsilon$. A *preconditioner* is commonly used to accelerate convergence [14]. In this paper, we refer to preconditioned CG as PCG. Gardner et al. [11] demonstrate that a modified version of PCG can be used to compute all terms in (1) and (2) simultaneously. This results in an algorithm for training and predicting with GPs that requires only a routine for performing matrix-vector products with the kernel matrix.

## 3   Method

To perform exact Gaussian process inference on large datasets, we must overcome the time and space requirements of solving linear systems. Most GP implementations use the Cholesky decomposition to solve linear systems required for inference [32]. The $\mathcal{O}(n^3)$ time complexity of the decomposition makes it difficult to perform exact GP inference on datasets with $n > 10^4$ data points without distributed computing and its associated communication overhead. In addition to this limitation, the Cholesky decomposition requires $\mathcal{O}(n^2)$ memory to store the lower-triangular factor $L$ in addition to the kernel matrix itself. At $n = 500{,}000$, the decomposition requires a full terabyte of memory and a prohibitively large amount of computational resources. Concurrent work by Nguyen et al. [26] was limited to exact GPs with $n \leq 120{,}000$ due to these drawbacks.

To address the above challenges, we build on Gardner et al. [11] and use preconditioned conjugate gradients (PCG) to solve linear systems. We overcome the memory limitations by partitioning the kernel matrix to perform all matrix-vector multiplications (MVMs) without ever forming the kernel matrix explicitly, reducing the memory requirement to $\mathcal{O}(n)$. In addition, we parallelize partitioned MVMs across multiple GPUs to further accelerate the computations, making training possible and timely even for datasets with $n > 10^6$.

$\mathcal{O}(n)$ **memory MVM-based inference.** The primary input to the modified PCG algorithm of Gardner et al. [11] is $\mathtt{mvm\_}\widehat{K}_{XX}$, a black-box function that performs MVMs using the kernel matrix $\widehat{K}_{XX}$.

Besides the storage cost associated with $\mathtt{mvm\_}\widehat{K}_{XX}$, each iteration of PCG updates four vectors: $\mathbf{u}$ (the current solution), $\mathbf{r}$ (the current error), $\mathbf{p}$ (the "search" direction for the next solution), and $\mathbf{z}$ (a preconditioned error term). Storing these vectors requires exactly $4n$ space. The quadratic space cost associated with PCG-based GP inference only comes from computing $\mathtt{mvm\_}\widehat{K}_{XX}$.

Typically in the full GP setting, $\mathtt{mvm\_}\widehat{K}_{XX}$ is implemented by first computing the full $n \times n$ kernel matrix $\widehat{K}_{XX}$, then computing the matrix-vector product with the full matrix. However, this would have the same $\mathcal{O}(n^2)$ memory requirement as Cholesky-based GP inference. Although forming $\widehat{K}_{XX}$ requires $\mathcal{O}(n^2)$ memory, the result of the MVM $\widehat{K}_{XX}\mathbf{v}$ requires only $\mathcal{O}(n)$ memory. Therefore, we reduce the memory requirement to $\mathcal{O}(n)$ by computing $\widehat{K}_{XX}\mathbf{v}$ in separate constant-sized pieces.

**Partitioned kernel MVMs.** To compute $\widehat{K}_{XX}\mathbf{v}$ in pieces, we partition the kernel matrix $\widehat{K}_{XX}$ such that *we only store a constant number of rows at any given time*. With the $4n$ memory requirement of storing the PCG vectors, our approach requires only $\mathcal{O}(n)$ memory.

We first partition the data matrix with $n$ points in $d$ dimensions, $X \in \mathbb{R}^{n \times d}$, into $p$ partitions each of which contains roughly $n/p$ data points:

$$X = \begin{bmatrix} X^{(1)}; & \cdots & ; X^{(p)} \end{bmatrix}$$

where we use ";" to denote row-wise concatenation. For each $X^{(l)}$, we can compute $\widehat{K}_{X^{(l)}X}$, which is a roughly $(n/p) \times n$ kernel matrix between the partition $X^{(l)}$ and the full data $X$. By partitioning

the kernel matrix this way, we rewrite it as a concatenation of the $p$ partitions:

$$\widehat{K}_{XX} = \left[\widehat{K}_{X^{(1)}X}; \quad \cdots \quad ; \widehat{K}_{X^{(p)}X}\right].$$

Computing each partition requires access to the full training set $X$, which we assume fits in memory. However, each partition $\widehat{K}_{X^{(l)}X}$ contains only $1/p$ of the entries of the full kernel matrix. Rewriting the matrix-vector product $\widehat{K}_{XX}\mathbf{v}$ in terms of these partitions,

$$\widehat{K}_{XX}\mathbf{v} = \left[\widehat{K}_{X^{(1)}X}\mathbf{v}; \quad \cdots \quad ; \widehat{K}_{X^{(p)}X}\mathbf{v}\right],$$

we see that this matrix-vector product can be computed in smaller components by separately computing each $\widehat{K}_{X^{(l)}X}\mathbf{v}$ and concatenating the results. We discard each kernel partition $\widehat{K}_{X^{(l)}X}$ once its MVM has been computed. This partitioning requires access to the training data $X$ and vector $\mathbf{v}$ already in memory and only allocates new memory to temporarily store the output vector $\mathbf{z}$ and a $(n/p) \times n$ kernel matrix partition $\widehat{K}_{X^{(l)}X}$. This algorithm allows us to reduce memory usage in exchange for sequential but easily parallelizable computations. If $p = 1$ then we have the naïve $\mathcal{O}(n^2)$ memory MVM procedure. As $p \to n$, PCG will only require $O(n)$ memory. In practice, we set a constant number of rows per partition according to the amount of memory available rather than number of partitions $p$. By keeping a partition in memory only until its component of the MVM has been computed, we can train GPs with an $\mathcal{O}(n)$ memory requirement.

**Distributed MVMs in Parallel.** MVM-based inference can easily take advantage of multiple GPUs or distributed computational resources because each MVM $\widehat{K}_{X^{(l)}X}\mathbf{v}$ can be performed on a different device. Thus we can compute multiple such MVMs in parallel to attain wall-clock speedups proportional to the number of devices available on large data sets where computation time exceeds the distributed computing overhead. Although $O(n)$ memory is achievable by setting $p = \mathcal{O}(n)$, in practice one may prefer $\mathcal{O}(n^2/p)$ memory to more effectively accelerate MVMs on parallel hardware with the necessary memory resources.

Additionally, we note that distributed parallel MVMs require only $O(n)$ communication. Each partitioned matrix multiplication only has to supply each device with a new right-hand-side vector $\mathbf{v}$. Finally, if $w$ devices are used, the output from each device will be a vector of length $n/pw$. Thus only $\mathcal{O}(n)$ data are copied to or from the devices. In contrast, distributing the Cholesky decomposition across multiple devices would require $\mathcal{O}(n^2)$ communication [15] [26].

Distributed computations have been utilized for approximate GP inference through mixture-of-experts GP models [7]. Concurrent with the Cholesky-based approach by Nguyen et al. [26], our method is the first to parallelize *exact* Gaussian process inference through distributed computing.

**Predictions.** At inference time, we must compute the predictive mean and variance given in (3) and (4). Although the predictive mean contains a linear solve $\widehat{K}_{XX}^{-1}\mathbf{y}$, this solve depends only on the training data. The result of this solve can be stored in a linear vector $\mathbf{a}$ and used for subsequent predictions. Therefore, computing the predictive mean requires computing $\widehat{K}_{\mathbf{x}^*X}\mathbf{a}$. Because this equation involves no linear solves, it can be computed efficiently on a single GPU.

Similarly, a training data dependent cache can be computed for the predictive variance using the method of Pleiss et al. [28] with a satisfactorily tight convergence tolerance. On exact GPs, this approach affords $\mathcal{O}(n)$ predictive variance computations,[2] removing the need to perform a linear solve at test time. In practice, we observe that both the predictive mean and variance can be computed in less than a second on a single GPU, even if the full model required days to train on multiple GPUs. Because predictions are fast after these precomputations, we can afford to use more stringent criteria for CG convergence for these one-time precomputations.

**Preconditioning.** To accelerate the convergence of CG, Gardner et al. [11] introduced a preconditioner for $\widehat{K}_{XX}$ derived from its *partial pivoted Cholesky* decomposition. Preconditioning works by modifying CG to solve the related linear system $P^{-1}K_{XX}\mathbf{v} = P^{-1}\mathbf{y}$ instead of solving the original system $K_{XX}\mathbf{v} = \mathbf{y}$. These linear systems have the same solution $\mathbf{v}^*$. However, the number of CG iterations required depends on the eigenvalue distribution of $P^{-1}K_{XX}$ rather than that of $K_{XX}$.

Computing a rank $k$ pivoted Cholesky preconditioner requires only $k$ kernel matrix rows: an already $\mathcal{O}(n)$ space dependence. While each iteration of CG requires computing each kernel matrix partition

from scratch, the preconditioner is computed once before any iterations of CG are performed. Therefore, it can be efficient to increase the size of the preconditioner to an extent if it reduces the number of CG iterations. While in Gardner et al. [11] the preconditioner size is typically limited to under 20 by default, in our use case we found that preconditioners of up to size $k = 100$ provide a noticeable improvement to wall-clock speed for large datasets.

**PCG Convergence Criteria.** Importantly, Conjugate gradients is *not* an approximate method for performing linear solves. Rather, it is a method that consumes time to perform solves to a specified tolerance. If this tolerance is low, the solve is exact. Thus, it is analogous to using gradient descent to solve a convex optimization problem (and is in fact largely what is happening). This gives us the ability to investigate how the performance of exact GPs change with different degrees of convergence.

At test time, we find that an accurate solve $\widehat{K}_{XX}^{-1}\mathbf{y}$ (with tolerance $\epsilon \leq 0.01$) is critical for good predictive performance; we therefore find that GP predictions require exact solves. For hyperparameter training, however, we find that, interestingly, less strict convergence criteria suffice, and even a looser convergence criterion of up to $\epsilon = 1$ has little impact on final model performance. Given that predictions using our approach are highly efficient (see Table 2), it may be interesting to investigate alternative approximate methods for finding good hyperparameters, and then using the techniques in this paper for exact inference and predicitons.

## 4    Related Work

**MVM-based GP inference.** Conjugate gradients and related MVM-based algorithms [9, 13, 40] have been used in certain settings throughout the GP literature. However, these methods have typically been used when the kernel matrix is structured and affords fast matrix-vector multiplications. Cunningham et al. [3] note that CG reduces asymptotic complexity when the data lie on a regularly-spaced grid because the kernel matrix is structured and affords $\mathcal{O}(n \log n)$ MVMs. This idea was extended to multi-dimensional grids by Saatçi [34]. Wilson and Nickisch [42] introduce a general-purpose GP approximation specifically designed for CG. They combine a structured inducing point matrix with sparse interpolation for approximate kernel matrices with nearly-linear MVMs.

More recently, there has been a push to use MVM-based methods on exact GPs. Cutajar et al. [4] use conjugate gradients to train exact GPs on datasets with up to $50,000$ points. The authors investigate using off-the-shelf preconditioners and develop new ones based on inducing-point kernel approximations.

**Approximate GP methods.** There are several approximations to GP inference that require $\leq O(n^2)$ memory and scale to large datasets. Perhaps the most common class of approaches are *inducing point methods* [29, 37], which introduce a set of $m \ll n$ data points $Z$ to form a low-rank kernel approximation:

$$K_{XX} \approx K_{XZ}K_{ZZ}^{-1}K_{ZX}.$$

Training and predictions with this approximation take $\mathcal{O}(nm^2)$ time and $\mathcal{O}(nm)$ space. Here we highlight some notable variants of the basic approach, though it is by no means an exhaustive list – see [22] for a more thorough review. *Sparse Gaussian process regression* (SGPR) [39] selects the inducing points $Z$ through a regularized objective. *Structured kernel interpolation* (SKI) [42] and its variants [12] place the inducing points on a grid, in combination with sparse interpolation, for $\mathcal{O}(n + g(m))$ computations and memory, where $g(m) \approx m$. *Stochastic variational Gaussian processes* (SVGP) [16] introduce a set of variational parameters that can be optimized using minibatch training. Recent work has investigated how to scale up the number of inducing points using tensor decompositions [10, 18].

## 5    Results

We compare the performance of exact Gaussian processes against widely-used scalable GP approximation methods on a range of large-scale datasets from the UCI dataset repository [1]. Our experiments demonstrate that exact GPs: (1) outperform popular approximate GPs methods on nearly all benchmarking datasets in our study; (2) compute thousands of test-point predictions in less than a second, even when $n > 10^6$; (3) utilize all available data when making predictions, even when $n > 10^5$; and (4) achieve linear training speedups on large datasets by adding additional GPU devices.

**Baselines.** We compare against two scalable GP approximations: Sparse Gaussian Process Regression (SGPR) [23, 39], and Stochastic Variational Gaussian Processes (SVGP) [16]. We choose these methods due to their popularity and general applicability, enabling a comparison over a wide range of datasets. SGPR is an inducing point method where the inducing points are learned through a variational objective. We use $m = 512$ for SGPR and $m = 1{,}024$ for SVGP, which are common values used for these methods [24]. We later experiment with varying the number of inducing points.

**Experiment details.** We extend the GPyTorch library [11] to perform all experiments. Each dataset is randomly split into $4/9$ training, $2/9$ validating, and $3/9$ testing sets. We use the validation set for tuning parameters like the CG training tolerance. The data is whitened to be mean 0 and standard deviation 1 as measured by the training set. We use a constant prior mean and a Matérn 3/2 kernel. We benchmark GPs with shared lengthscales across the input dimension in Table 1 as well as GPs with independent lengthscales in the appendix.

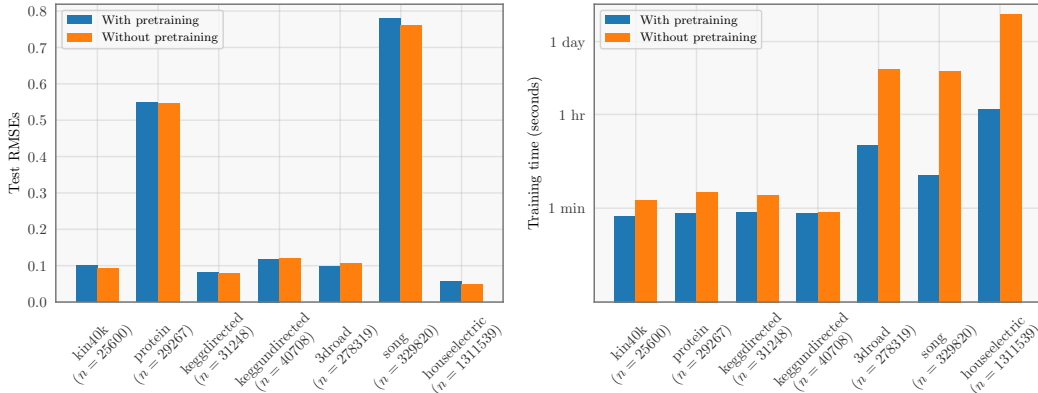

Figure 1: A comparison of exact GPs trained using our initialization procedure against exact GPs trained for 100 iterations using Adam. Better initializatoin allows exact GPs to achieve similar RMSEs while requiring drastically less training time on large datasets.

We learn model hyperparameters and variational parameters by optimizing the log marginal likelihood. For SGPR, we perform 100 iterations of Adam with a learning rate of 0.1. For SVGP, we perform 100 epochs of Adam with a minibatch size of 1,024 and a learning rate of 0.01, which we found to perform better than 0.1. *For exact GPs, the number of optimization steps has the greatest effect on the training time for large datasets.* To reduce the training time for exact GPs, we first randomly subset 10,000 training points from the full training set to fit an exact GP whose hyperparameters will be used as initialization. We pretrain on this subset with 10 steps of L-BFGS [21] and 10 steps of Adam [19] with 0.1 step size before using the learned hyperaparameters to take 3 steps of Adam on the full training dataset. Figure 1 shows that this initialization plus fine-tuning procedure achieves comparable test performance to running Adam for the full 100 iterations without pretraining. We do not pretrain the SGPR and SVGP models because we found that they required a significant number of fine-tuning steps after pretraining due to their increased number of model parameters. We show additional training statistics for exact GPs trained with 100 steps of Adam in the appendix.

For all experiments, we use a rank-100 partial pivoted-Cholesky preconditioner and run PCG with a tolerance of $\epsilon = 1$ during training. We constrain the learned noise to be at least 0.1 to regularize the poorly conditioned kernel matrix for the houseelectric dataset. We perform all training on a single machine with 8 NVIDIA Tesla V100-SXM2-32GB-LS GPUs. Code to reproduce the experiments is available at `https://gpytorch.ai`.

**Accuracy.** Table 1 displays the accuracy and negative log likelihoods of exact GPs and approximate methods on several large-scale datasets using a single lengthscale across dimensions. *The results for independent lengthscale GPs can be found in the appendix.* We find that exact GPs achieve lower error than approximate methods on nearly every dataset. Notably, on certain datasets like Kin40K and CTslice, exact GPs achieve a half or even a quarter of the error of some approximate methods, and consistently outperform the approximate methods even on datasets with up to over 1M data points. Although Nguyen et al. [26] show results for exact GPs on $n < 120{,}000$, this is the first set of results comparing exact GPs to approximate GPs on $n \gg 10^5$.

Table 1: Root-mean-square-error (RMSE) and negative log-likelihood (NLL) of exact GPs and approximate GPs on UCI regression datasets using a constant prior mean and a Matérn 3/2 kernel with a shared lengthscale across all dimensions. All trials were averaged over 3 trials with different splits. $n$ and $d$ are the size and dimensionality of the training dataset, respectively. The number of GPUs used and the number of kernel partions are reported in Table 2. We were unable to scale SGPR to HouseElectric due to its memory requirements when $m = 512$.

| | | | RMSE | | | NLL | | |
|---|---|---|---|---|---|---|---|---|
| Dataset | $n$ | $d$ | Exact GP (BBMM) | SGPR ($m=512$) | SVGP ($m=1,024$) | Exact GP (BBMM) | SGPR ($m=512$) | SVGP ($m=1,024$) |
| PoleTele | 9,600 | 26 | **0.151** ± 0.012 | 0.217 ± 0.002 | 0.215 ± 0.002 | **−0.180** ± 0.036 | −0.094 ± 0.008 | −0.001 ± 0.008 |
| Elevators | 10,623 | 18 | **0.394** ± 0.006 | 0.437 ± 0.018 | 0.399 ± 0.009 | 0.619 ± 0.054 | 0.580 ± 0.060 | **0.519** ± 0.022 |
| Bike | 11,122 | 17 | **0.220** ± 0.002 | 0.362 ± 0.004 | 0.303 ± 0.004 | **0.119** ± 0.044 | 0.291 ± 0.032 | 0.272 ± 0.018 |
| Kin40K | 25,600 | 8 | **0.099** ± 0.001 | 0.273 ± 0.025 | 0.268 ± 0.022 | **−0.258** ± 0.084 | 0.087 ± 0.067 | 0.236 ± 0.077 |
| Protein | 29,267 | 9 | **0.536** ± 0.012 | 0.656 ± 0.010 | 0.668 ± 0.005 | 1.018 ± 0.056 | **0.970** ± 0.010 | 1.035 ± 0.006 |
| KeggDirected | 31,248 | 20 | **0.086** ± 0.005 | 0.104 ± 0.003 | 0.096 ± 0.001 | −0.199 ± 0.381 | **−1.123** ± 0.016 | −0.940 ± 0.020 |
| CTslice | 34,240 | 385 | 0.262 ± 0.448 | **0.218** ± 0.011 | 1.003 ± 0.005 | **−0.894** ± 0.188 | −0.073 ± 0.097 | 1.422 ± 0.005 |
| KEGGU | 40,708 | 27 | **0.118** ± 0.000 | 0.130 ± 0.001 | 0.124 ± 0.002 | −0.419 ± 0.027 | **−0.984** ± 0.012 | −0.666 ± 0.007 |
| 3DRoad | 278,319 | 3 | **0.101** ± 0.007 | 0.661 ± 0.010 | 0.481 ± 0.002 | 0.909 ± 0.001 | 0.943 ± 0.002 | **0.697** ± 0.002 |
| Song | 329,820 | 90 | 0.807 ± 0.024 | **0.803** ± 0.002 | 0.998 ± 0.000 | **1.206** ± 0.024 | 1.213 ± 0.003 | 1.417 ± 0.000 |
| Buzz | 373,280 | 77 | **0.288** ± 0.018 | 0.300 ± 0.004 | 0.304 ± 0.012 | 0.267 ± 0.028 | **0.106** ± 0.008 | 0.224 ± 0.050 |
| HouseElectric | 1,311,539 | 9 | **0.055** ± 0.000 | —— | 0.084 ± 0.005 | **−0.152** ± 0.001 | —— | −1.010 ± 0.039 |

Table 2: Timing results for training and prediction for exact GPs and approximate GPs. Training times were recorded using the same hardware and other experimental details as in Table 1. Except for †, all trials were averaged over 3 trials with different splits. $p$ is the number of kernel partitions used to train the exact GP. Prediction times were measured by computing 1,000 predictive means and variances on 1 NVIDIA RTX 2080 Ti GPU An asterisk (*) indicates the one-time pre-computed cache was calculated using 8 V100 GPUs. Best results are in bold (lower is better).

| | Training | | | | | Precomputation | Prediction | | |
|---|---|---|---|---|---|---|---|---|---|
| Dataset | Exact GP (BBMM) | SGPR ($m=512$) | SVGP ($m=1,024$) | #GPUs | $p$ | Exact GP (BBMM) | Exact GP (BBMM) | SGPR ($m=512$) | SVGP ($m=1,024$) |
| PoleTele | **41.5s** ± 1.1 | 69.5s ± 20.5 | 68.7s ± 4.1 | 1 | 1 | 5.14 s | **6 ms** | **6 ms** | 273 ms |
| Elevators | **41.0s** ± 0.7 | 69.7s ± 22.5 | 76.5s ± 5.5 | 1 | 1 | 0.95 s | **7 ms** | **7 ms** | 212 ms |
| Bike | **41.2s** ± 0.9 | 70.0s ± 22.9 | 77.1s ± 5.6 | 1 | 1 | 0.38 s | **7 ms** | 9 ms | 182 ms |
| Kin40K | **42.7s** ± 2.7 | 97.3s ± 57.9 | 195.4s ± 14.0 | 1 | 1 | 12.3 s | **11 ms** | 12 ms | 220 ms |
| Protein | **47.9s** ± 10.1 | 136.5s ± 53.8 | 198.3s ± 15.9 | 1 | 1 | 7.53 s | 14 ms | **9 ms** | 146 ms |
| KeggDirected | **51.0s** ± 6.3 | 132.0s ± 65.6 | 228.2s ± 22.9 | 1 | 1 | 8.06 s | **15 ms** | 16 ms | 143 ms |
| CTslice | 199.0s ± 299.9 | **129.6s** ± 59.2 | 232.1s ± 20.5 | 1 | 1 | 7.57 s | 22 ms | 14 ms | 133 ms |
| KEGGU | **47.4s** ± 8.6 | 133.4s ± 62.7 | 287.0s ± 24.1 | 8 | 1 | 18.9 s | 18 ms | 13 ms | 211 ms |
| 3DRoad | 947.8s ± 443.8 | **720.5s** ± 330.4 | 2045.1s ± 191.4 | 8 | 16 | 118 m* | 119 ms | **68 ms** | 130 ms |
| Song | **253.4s** ± 221.7 | 473.3s ± 187.5 | 2373.3s ± 184.9 | 8 | 16 | 22.2 m* | 123 ms | **99 ms** | 134 ms |
| Buzz | 4283.6s ± 1407.2 | **1754.8s** ± 1099.6 | 2780.8s ± 175.6 | 8 | 19 | 42.6 m* | 131 ms | **114 ms** | 142 ms |
| HouseElectric | **4317.3s** ± 147.2 | —— | 22062.6s ± 282.0 | 8 | 218 | 3.40 hr* | 958 ms | —— | **166 ms** |

Interestingly, the size or the dimensionality of the dataset does not seem to influence the relative performance of the approximate methods. For example, though Protein and Kin40K are similar in size and have almost the same dimensionality, the approximate methods perform worse on Kin40K (relative to the RMSE of exact GPs). Moreover, we also see that the choice of approximate method affects performance, with neither approximate method consistently outperforming the other.

**Training time.** Table 2 displays the training time for exact and approximate GPs. On datasets with $n < 35,000$, an exact GP fits on a single 32GB GPU without any partitioning and can be trained in less than a minute. For $n \geq 100,000$, we must use kernel partitioning as discussed above, which significantly increases the training time for exact GPs. However, if the necessary computational resources are available, these experiments show that it may be preferable to train an exact GP to make more accurate predictions in exchange for longer training times.

**Training acceleration with multiple GPUs.** Because we use matrix-multiplication-based approaches to train exact GPs, the computations can be easily parallelized and distributed. Moreover, matrix multiplication is one of the most commonly distributed routines, so parallelized GP implementations can be built using readily-available routines in libraries like PyTorch [27]. Figure 2 plots the speedup as more GPUs are used for training on the KEGGU, 3DRoad, Song, and Buzz datasets. Each of these datasets achieve a nearly linear speedup when adding up to 4 GPUs. The speedup is more pronounced for the two large datasets (3DRoad and Song) that require kernel partitioning. The training time can be further improved by using more GPUs to reduce the number of kernel partitions.

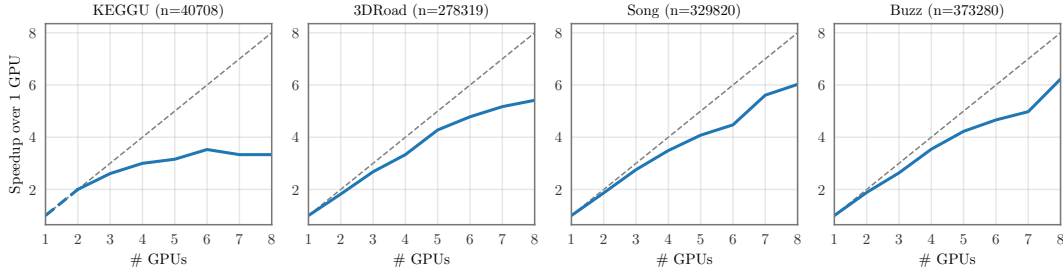

Figure 2: Training speedup from using additional GPUs at training time. Since our exact GPs use matrix-multiplication based inference, they achieve a near linear speedup with more computing resources on large datasets.

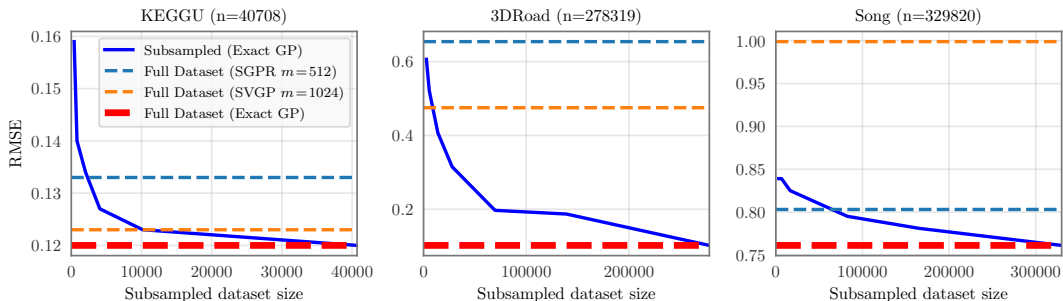

Figure 4: Test root-mean-square error (RMSE) as a function of subsampled dataset size (lower is better). Subsampled exact GPs outperform approximate GPs even with a quarter of the training set. Exact GP error continues to decrease as data is added until the full dataset is used.

**Prediction time.** Although exact GPs take longer to train, we find that *their speed is comparable to approximate methods at test time.* After training various GPs, we follow the common practice of precomputing the work required for GP predictions [28].

Table 2 displays the time to compute 1,000 predictive means and variances at test time before and after precomputation. All predictions are made on one NVIDIA RTX 2080 Ti GPU. We see exact GPs take *less than a second for all predictions* across all dataset sizes used.

## 5.1 Ablation Studies

With our method, we can better understand how exact GPs and approximate GPs scale to datasets with $n \gg 10^4$. Here, we demonstrate how the amount of data affects exact GP performance, and how the number of inducing points affects the performance of approximate GPs.

**Do GPs need the entire dataset?** As a non-parametric model, Gaussian processes naturally adapt to the amount of training data available [44]. Figure 4 shows an increase in accuracy as we increase the amount of training data on the KEGGU, 3DRoad, and Song datasets. For each dataset, we subsample a fraction of the training data and plot the resulting root-mean-square error on the test-set as a function of subsampled training set size. We use the same 1/3 holdout of the full dataset to test in each case. As expected, the test RMSE decreases monotonically as we increase the subsample size. Figure 4 also

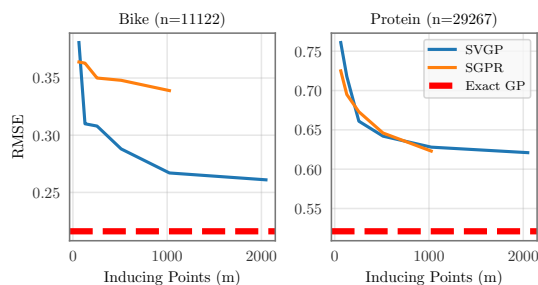

Figure 3: Error of SVGP and SGPR methods as a function of the number of inducing points ($m$). Both methods scale cubically with $m$. We were unable to run SGPR with more than 1,024 inducing points on a single GPU. Exact GPs have lower error than both methods.

shows the performance of exact GP, SGPR, and SVGP trained on the entire training set. Strikingly, in

all three cases, *an exact GP with less than a quarter of the training data outperformed approximate GPs trained on the entire training set*. Furthermore, test error continues to decrease with each addition of training data.

**Would more inducing points help?** In Table 1, exact GPs uniformly take substantially longer to train on the largest datasets than the approximate methods. This naturally raises the question "can approximate models with more inducing points recover the performance of exact methods?" We plot test RMSE on two datasets, Bike and Protein, as a function of the number of inducing points in Figure 3. The test RMSE of both inducing point methods saturates on both datasets well above the test RMSE of an exact GP. Furthermore, we note that using $m$ inducing points introduces a $m \times m$ matrix and a $\mathcal{O}(nm^2 + m^3)$ time complexity [16, 17] which makes it difficult to train SGPR with $m \gg 1024$ inducing points on one GPU. It is possible to combine kernel partitioning with inducing-point methods to utilize even larger values of $m$. However, as Figure 3 and Table 1 show, it may be preferable to use the extra computational resources to train an exact GP on more data rather than to train an approximate GP with more inducing points.

# 6 Discussion

Historically, for Gaussian processes, "a large dataset is one that contains over a few thousand data points" [16] which have traditionally necessitated scalable approximations. Bigger datasets have traditionally necessitated scalable approximations. In this paper, we have extended the applicability of exact GPs far beyond what has been thought possible — to datasets with over a million training examples through MVM-based GP inference. Our approach uses easily parallelizable routines that fully exploit modern parallel hardware and distributed computing. In our comparisons, we find that exact GPs are more widely applicable than previously thought, performing significantly better than approximate methods on large datasets, while requiring fewer design choices.

**Is CG still exact?** In the GP literature, *exact* GP inference typically refers to using the Cholesky decomposition with exact kernels [32]. A natural question to ask is whether we can consider our approach "exact" in light of the fact that CG perform solves only up to a pre-specified error tolerance. However, unlike the approximate methods presented in this paper, the difference between a CG-based model and a theoretical model with "perfect" solves can be precisely controlled by this error tolerance. We therefore consider CG exact in a sense that is commonly used in the context of mathematical optimization — namely that it computes solutions up to arbitrary precision. In fact, CG-based methods can often be more precise than Cholesky based approaches in floating-point arithmetic due to fewer round-off errors [11].

**When to approximate?** There are many approximate methods for scalable Gaussian processes, with varying statistical properties, advantages, and application regimes. We chose to compare exact GPs to approximation methods SVGP and SGPR for their popularity and available GPU implementations. There may be some regimes where other approximate methods or combinations of methods outperform these two approximations. Our objective is not to perform an exhaustive study of approximate methods and their relative strengths but to highlight that such comparisons are now possible with modern hardware.

Indeed, there are cases where an approximate GP method might still be preferable. Examples may include training on large datasets with limited computational resources. In certain regimes, such as low dimensional spaces, there are approximations that are designed to achieve high degrees of accuracy in less time than exact GPs. Additionally, GP inference with non-Gaussian likelihoods (such as for classification) requires an approximate inference strategy. Some approximate inference methods, such as Laplace and MCMC [25, 32], may be amenable to the parallelisation approaches discussed here for approximate inference with exact kernels.

Nonetheless, with efficient utilization of modern hardware, exact Gaussian processes are now an appealing option on substantially larger datasets than previously thought possible. Exact GPs are powerful yet simple – achieving remarkable accuracy without requiring much expert intervention. We expect exact GPs to become ever more scalable and accessible with continued advances in hardware design.

**Acknowledgments**

KAW and AGW are supported by NSF IIS-1910266, NSF IIS-1563887, Facebook Research, NSF I-DISRE 1934714, and an Amazon Research Award. GP and KQW are supported in part by the III-1618134, III-1526012, IIS-1149882, IIS-1724282, and TRIPODS-1740822 grants from the National Science Foundation. In addition, they are supported by the Bill and Melinda Gates Foundation, the Office of Naval Research, and SAP America Inc.

## Footnotes

[2]We do not achieve constant-time predictions as described in [28]. Reducing the $\mathcal{O}(n)$ prediction time to $\mathcal{O}(1)$ requires using structure kernel interpolation [42] to approximate the kernel matrix.

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
