[Supplementary Material]

# Appendix to Exact Gaussian Processes on a Million Data Points

**Exact GPs with independent lengthscales**  Table 1 shows the performance of exact GPs against approximate methods when using independent lengthscales across the input dimension, which is frequently used in practice to attain better accuracy [1]. Here, we also see that exact GPs are generally more accurate than SGPR and SVGP. Table 2 shows the corresponding training time for these experiments.

Table 1: Exact GPs vs approximate GPs on medium and large regression datasets with an independent lengthscale per dimension. All experiments were averaged over 3 different splits using the same experimental setup as in the main paper with pretraining.

| | | | RMSE | | | NLL | | |
|---|---|---|---|---|---|---|---|---|
| Dataset | $n$ | $d$ | Exact GP (BBMM) | SGPR ($m=512$) | SVGP ($m=1,024$) | Exact GP (BBMM) | SGPR ($m=512$) | SVGP ($m=1,024$) |
| PoleTele | 9,600 | 26 | $\mathbf{0.088} \pm 0.003$ | $0.113 \pm 0.005$ | $0.109 \pm 0.002$ | $-0.660 \pm 0.081$ | $-0.817 \pm 0.005$ | $\mathbf{-0.644} \pm 0.008$ |
| Elevators | 10,623 | 18 | $0.399 \pm 0.011$ | $0.426 \pm 0.007$ | $\mathbf{0.388} \pm 0.010$ | $0.626 \pm 0.043$ | $0.528 \pm 0.015$ | $\mathbf{0.486} \pm 0.019$ |
| Bike | 11,122 | 17 | $\mathbf{0.043} \pm 0.012$ | $0.094 \pm 0.010$ | $0.077 \pm 0.005$ | $\mathbf{-1.323} \pm 0.170$ | $-0.805 \pm 0.005$ | $-0.984 \pm 0.021$ |
| Kin40K | 25,600 | 8 | $\mathbf{0.080} \pm 0.001$ | $0.225 \pm 0.026$ | $0.240 \pm 0.007$ | $\mathbf{-0.755} \pm 0.009$ | $-0.073 \pm 0.055$ | $0.091 \pm 0.033$ |
| Protein | 29,267 | 9 | $\mathbf{0.511} \pm 0.009$ | $0.619 \pm 0.003$ | $0.613 \pm 0.011$ | $0.960 \pm 0.033$ | $\mathbf{0.915} \pm 0.004$ | $0.952 \pm 0.018$ |
| KeggDirected | 31,248 | 20 | $\mathbf{0.083} \pm 0.001$ | $0.104 \pm 0.002$ | $0.105 \pm 0.003$ | $-0.838 \pm 0.031$ | $\mathbf{-1.163} \pm 0.005$ | $-0.853 \pm 0.033$ |
| CTslice | 34,240 | 385 | $0.497 \pm 0.029$ | $\mathbf{0.217} \pm 0.009$ | $1.004 \pm 0.005$ | $0.939 \pm 0.004$ | $\mathbf{-0.037} \pm 0.060$ | $1.423 \pm 0.005$ |
| KEGGU | 40,708 | 27 | $\mathbf{0.120} \pm 0.001$ | $0.130 \pm 0.001$ | $0.126 \pm 0.002$ | $-0.540 \pm 0.035$ | $\mathbf{-1.049} \pm 0.010$ | $-0.653 \pm 0.013$ |
| 3DRoad | 278,319 | 3 | $\mathbf{0.110} \pm 0.017$ | $0.578 \pm 0.001$ | $0.390 \pm 0.005$ | $1.239 \pm 0.025$ | $0.791 \pm 0.033$ | $\mathbf{0.486} \pm 0.010$ |
| Song | 329,820 | 90 | $\mathbf{0.774} \pm 0.001$ | $0.816 \pm 0.038$ | $0.998 \pm 0.000$ | $\mathbf{1.162} \pm 0.002$ | $1.243 \pm 0.083$ | $1.417 \pm 0.000$ |
| Buzz | 373,280 | 77 | $0.279 \pm 0.002$ | $0.289 \pm 0.001$ | $\mathbf{0.270} \pm 0.012$ | $0.161 \pm 0.026$ | $\mathbf{0.092} \pm 0.017$ | $0.119 \pm 0.042$ |
| HouseElectric | 1,311,539 | 9 | $\mathbf{0.054} \pm 0.000$ | —— | $0.127 \pm 0.046$ | $\mathbf{-0.207} \pm 0.001$ | —— | $0.024 \pm 0.984$ |

Table 2: Exact GPs vs approximate GPs on medium and large regression datasets with an independent lengthscale per dimension.

| | Training Time (s) | | | | |
|---|---|---|---|---|---|
| Dataset | Exact GP (BBMM) | SGPR ($m=512$) | SVGP ($m=1,024$) | #GPU | $p$ |
| PoleTele | $\mathbf{40.8s} \pm 0.0$ | $71.6s \pm 23.7$ | $67.6s \pm 5.6$ | 1 | 1 |
| Elevators | $\mathbf{40.5s} \pm 0.0$ | $72.4s \pm 26.6$ | $76.1s \pm 4.0$ | 1 | 1 |
| Bike | $\mathbf{40.6s} \pm 0.1$ | $72.1s \pm 25.0$ | $75.5s \pm 4.5$ | 1 | 1 |
| Kin40K | $\mathbf{41.2s} \pm 0.0$ | $92.5s \pm 50.1$ | $184.8s \pm 17.1$ | 1 | 1 |
| Protein | $\mathbf{42.3s} \pm 0.1$ | $136.4s \pm 49.1$ | $199.7s \pm 15.8$ | 1 | 1 |
| KeggDirected | $\mathbf{46.7s} \pm 4.5$ | $171.2s \pm 31.5$ | $219.9s \pm 17.8$ | 1 | 1 |
| CTslice | $\mathbf{41.7s} \pm 0.0$ | $133.7s \pm 53.6$ | $230.0s \pm 18.0$ | 1 | 1 |
| KEGGU | $\mathbf{42.3s} \pm 0.2$ | $142.7s \pm 59.6$ | $285.6s \pm 22.1$ | 8 | 1 |
| 3DRoad | $3592.5s \pm 9.4$ | $\mathbf{545.0s} \pm 60.6$ | $2035.9s \pm 185.4$ | 8 | 16 |
| Song | $\mathbf{139.3s} \pm 0.6$ | $445.7s \pm 170.5$ | $2373.7s \pm 167.4$ | 8 | 16 |
| Buzz | $\mathbf{1827.0s} \pm 379.3$ | $1899.1s \pm 164.6$ | $2780.4s \pm 191.4$ | 8 | 19 |
| HouseElectric | $\mathbf{5563.1s} \pm 51.0$ | —— | $11982.2s \pm 455.2$ | 8 | 218 |

**Exact GPs with 100 steps of Adam** Although we used pretraining and finetuning in our main experiments to reduce training time, we also trained exact GPs using 100 steps of Adam to ensure a fair comparison aginst SGPR and SVGPs that were trained with Adam. Table 3 shows this comparison. Furthermore, we found that it is sometimes unnecessary to take 100 iterations of Adam on large datasets, shown in Figure 1. Thus when training exact GPs on large datasets, we are able to take fewer optimization steps to reduce the training time without sacrificing accuracy.

Table 3: Exact GPs vs approximate GPs on medium and large regression datasets with a shared lengthscale per dimension trained using 100 steps of Adam with 0.1 step size.

| | | | RMSE (random = 1) | | | Training Time | | | | |
| Dataset | $n$ | $d$ | Exact GP (BBMM) | SGPR ($m=512$) | SVGP ($m=1,024$) | Exact GP (BBMM) | SGPR ($m=512$) | SVGP ($m=1,024$) | #GPU | $p$ |
|---|---|---|---|---|---|---|---|---|---|---|
| PoleTele | 9,600 | 26 | **0.154** | 0.219 | 0.218 | **22.1 s** | 40.6 s | 68.1 s | 1 | 1 |
| Elevators | 10,623 | 18 | **0.374** | 0.436 | 0.386 | **17.1 s** | 41.2 s | 112 s | 1 | 1 |
| Bike | 11,122 | 17 | **0.216** | 0.345 | 0.261 | **18.8 s** | 41.0 s | 109 s | 1 | 1 |
| Kin40K | 25,600 | 8 | **0.093** | 0.257 | 0.177 | 83.3 s | **56.1 s** | 297 s | 1 | 1 |
| Protein | 29,267 | 9 | **0.545** | 0.659 | 0.640 | 120 s | **65.5 s** | 300 s | 1 | 1 |
| KeggDirected | 31,248 | 20 | **0.078** | 0.089 | 0.083 | 107 s | **67.0 s** | 345 s | 1 | 1 |
| CTslice | 34,240 | 385 | **0.050** | 0.199 | 1.011 | 148 s | **77.5 s** | 137 s | 1 | 1 |
| KEGGU | 40,708 | 27 | **0.120** | 0.133 | 0.123 | **50.8 s** | 84.9 s | 7.61 min | 8 | 1 |
| 3DRoad | 278,319 | 3 | **0.106** | 0.654 | 0.475 | 7.06 hr | **8.53 min** | 22.1 min | 8 | 16 |
| Song | 329,820 | 90 | **0.761** | 0.803 | 0.999 | 6.63 hr | **9.38 min** | 18.5 min | 8 | 16 |
| Buzz | 373,280 | 77 | **0.265** | 0.387 | 0.270 | 11.5 hr | **11.5 min** | 1.19 hr | 8 | 19 |
| HouseElectric | 1,311,539 | 9 | **0.049** | —— | 0.086 | 3.29 days | —— | **4.22 hr** | 8 | 218 |

Figure 1: Figure comparing an exact GP trained with 100 steps of Adam against an exact GP trained partially with Adam.

# References

[1] C. E. Rasmussen and C. K. Williams. *Gaussian processes for machine learning*, volume 1. MIT press Cambridge, 2006.