[Reviews · NeurIPS 2019]

Reviewer 1



*** Update after the rebuttal *** I thank the authors for their clarifications and additional results. A good result from this rebuttal exercise is that the authors now report test likelihoods, which I think should always be reported in GP papers. It will be great if the authors can be a bit more explicit about the additional contributions with respect to the previous work [11,26]. It is OK to focus on the engineering effort and experimental evaluation as the main contribution of the paper. The paper will also benefit from reporting error bars on all the results. **** This paper develops a method to scale up exact GP regression (under the standard Gaussian noise assumption) to a very large number of observations. The paper does not provide new theoretical developments or a technical contribution. The implemented method is based on the work by Gardner et al [11] and by Pleis et al [25]. The paper is well written and structured and it is clear in terms of the claimed contributions, i.e. using the previous developments to provide a practical implementation that can scale up GP regression. From this perspective, there is not much to comment on the technical part as there are not really new insights here. From the practical perspective, the paper provides two interesting conclusions: (i) that for the benchmarks studied, exact GP regression can indeed benefit from using all data and (ii) inducing-variable approximations are somewhat limited by the number of inducing variables and by the main assumption in the approximate posterior. In terms of clarity, it would be helpful if the authors can clarify the differences between SGPR and SVGP (line 220) as both methods reference 16. ### Significance The main question is whether these practical contributions are sufficient to deserve publication at NeuriPS. Although I find the above insights interesting, I guess perhaps NeurIPS is not the best venue for this work. There is also one major flaw in the evaluation of the method: most GP researchers (and certainly the NeurIPS community) are interested in the full predictive distribution so I believe a thorough evaluation of the predictive variances must also be presented. These days, the ML community has plenty of flexible non-probabilistic supervised learning methods and those are not even mentioned here.

Reviewer 2



-- Paper Summary -- Exact training and inference with Gaussian processes (GPs) is characterised by cubic time complexity and squared memory requirements, rendering their application to very large datasets difficult or even impossible. This concern has been widely addressed in the literature by developing approximations to standard GPs, with techniques varying from inducing points, structure-exploring mechanisms, etc. In this paper, the authors argue that available computational power should instead be directed towards preserving the exactness of the original GP model, and develop a scheme that exploits the efficiency of iterative solvers implemented on GPUs for attaining this goal. This scheme enables the authors to show results for datasets having well over a few thousands of data points (up to one million), which is well beyond the scope to which exact GP training/inference was considered feasible. This also makes it possible to carry out a more direct comparison to GP approximations applied to such large datasets, which in turn corroborates the authors’ hypothesis that exact GPs consistently perform better than approximations in a wide variety of settings. -- Writing/Clarity -- The paper is well-written and I greatly enjoyed reading it. The exposition is clear, and the methodology is easy to follow. However, I am not convinced that the Related Work section works best at the end for this paper. Given that much of the work is the realisation of several concepts discussed in preceding work (but admittedly not properly implemented or sufficiently explored), I believe it would be more appropriate to highlight this work earlier on. A few further comments: - Section numbering is somehow missing. - In equations (1) and (2), $^{-1}$ was left out for some of the kernel matrices. - I would include more technical details in an appendix, especially with regards to the preconditioning scheme used and the general (preconditioned) conjugate gradient algorithm. The latter would give greater meaning to the elements listed in L105. - Table 2 is strangely positioned in the middle of the text. Move to top or bottom of the page. - References need to be updated properly in order to ensure words such as Gaussian, Cholesky, etc are capitalised. - Some minor spotted typos: L120: Missing punctuation for equation; L131: rather ‘than’; L133: use train -> train; L189: multiplies -> multiplications; L286: Author name can be omitted; L310: ‘trained’ is unnecessary; -- Originality and Significance -- I consider this to be a very timely topic, especially with the increased interest in using GPs for more large-scale practical problems which had been typically addressed using neural networks in the past. In recent years, the literature on GPs has been predominantly focused on developing improved approximations, either based on inducing points, exploiting structure, etc. Indeed, many extensions to the plain GP model (for example convolutional or recurrent GPs) are also predominantly based on approximations relying on inducing points in order to enable scalability to larger datasets. On the other hand, the motivation for preserving the exactness of a GP in this work is more aligned with the findings of the cited work by Cutajar et al (2016), whose results also indicate that exact GP training and inference should be preferred to standard approximations (albeit on a much smaller scale than the datasets considered in this paper). Very recently published work on distributed Cholesky deserves further attention however, since this could be considered in direct comparison to the methodology featured here: ‘Exact Gaussian Process Regression with Distributed Computations’, Nguyen et al (2019). The aforementioned paper shares several goals with those presented in this paper, and it would be helpful for the authors to compare against the contributions and outcomes featured in that work. A practical comparison may be impractical unless code is already available for their method, but some commentary on the expected complexity should definitely be included. Indeed, the notion of distributing the Cholesky decomposition itself is dismissed quite quickly in this submission, which is why the Related Work section in the indicated paper could be used to expand on the provided discussion. Several of the results presented in this paper are particularly noteworthy, such as the experiment showing how using an exact GP on a portion of the available data is still preferable to an approximation applied to the full dataset. Such a result should be appropriately publicised in the GP community, and such a paper could have implications on whether more effort will be directed towards improving existing approximations, or rather building new GP extensions on the exact framework championed by this paper. On the flip-side, one could also argue that the contributions themselves owe more to a committed ‘engineering’ effort rather than theoretical innovation. Indeed, the notion of partitioning the kernel matrix when computing matrix-vector products (which is the primary enabler of the speed improvements detailed here) has already been hypothesised or explored in earlier work. Yet, to the best of my knowledge, this is the first time this has been considered in the context of GPs on modern GPU architectures, as a result of which the scale of datasets considered here is also well beyond that considered elsewhere. In view of the above, I believe that the paper’s results are more interesting that the technical contributions themselves, which are understandably more implementation-dependent. For this reason, I would have appreciated more experiments in the vein of those featured in the ‘Ablation studies’ section. This paper could address several important points on why exact GP regression should be favoured over approximations, but I think that such a discussion is currently muted in both the introductory and concluding remarks. In the introduction, the paper is positioned as a ‘signpost’ for ‘theorists to propose even better approximations to address this gap in performance [with standard GP approximations]’, whereas I think the results presented in this paper have stronger implications for carrying out more direct comparisons against techniques such as neural networks. Don’t the outcomes already imply that exact GP regression is the way forward? Meanwhile, in the conclusion, it is stated that ‘There are still cases where approximate GP methods are preferable’, but I don’t think this statement is sufficiently substantiated in the preceding sections. On a similar note, there is an interesting comment on L183/184 about how approximations may be just as good for the training procedure - I believe this warrants better context and discussion than is currently provided. Another weakness of this paper is its isolation to the plain GP regression setting. Although this is expected given the methodology used to enable tractability, I would have appreciated at least some discussion into whether any of the material presented here can be extended to the classification setting. Of course, one could argue that using Laplace or EP already implicitly takes away from the ‘exactness’ of a GP, but I think there is still scope for having an interesting discussion here (possibly akin to that provided in Cutajar et al, 2015). Likewise, any intuition of how/whether this can be extended to more advanced GP set-ups, such as multi-task, convolutional, and recurrent variations (among many others) would also be useful. -- Technical Quality/Evaluation -- The technical contributions and implementation details are easy to follow, and I did not find any faults in that respect. The experimental evaluation is also varied and convincingly shows that exact GP inference widely outperforms standard approximations. Nonetheless I have a few concerns listed below; - It appears that in nearly all experiments, the results are reported for a single held-out test set. Standard practice in most papers on GPs involves using a number of train/test splits or folds which give a more accurate illustration of the method’s performance. While I imagine that the size of the datasets considered in this work entail that this can take quite a long time to complete, I highly encourage the authors to carry out this exercise; - If I understood correctly, a kernel with a shared length scale is used in all experiments, which does not conform to the ARD kernels one would typically use in a practical setting. While several papers presenting approximate GPs have also made this assumption in the past (e.g. Hensman et al (2013)), more recent work such as the AutoGP by Krauth et al. (2017) emphasise why ARD should more consistently be used, and demonstrate how automatic differentiation frameworks minimise the performance penalty introduced by using such schemes. I believe this has to be addressed more clearly in the paper, and would also give more meaning to the commentary provided in L243-248, which otherwise feels spurious. I consider this to be crucial for painting a more complete picture in the Results section. - For a GP paper, I also find it strange that results for negative log likelihood are not reported here. While these are expected to follow a similar trend to RMSE, I would definitely include such results in a future version of the manuscript since this also has implications on uncertainty calibration. On a related note, I was surprised this paper did not have any supplementary material attached, because further experiments and more results would definitely be welcome. -- Overall recommendation -- This paper does not introduce any major theoretical elements, but the technical contributions featured here, along with the associated practical considerations, are timely in showing how modern GPU architectures can be exploited for carrying out exact GP training and inference to an extent which had not previously been considered. The paper is well written and some of the results should indeed stimulate interesting discussions on whether standard GP approximations are still worthwhile for plain regression tasks. Unfortunately, given how this paper’s worth relies heavily on the quality of the experimental results, there are a few technical issues in the paper which I believe should be addressed in a published version. I also think that several of the discussions featured in the paper can be expanded further - the authors should not refrain from including their own intuition on the broader implications of this work, which I feel is currently missing. -- Post-rebuttal update -- I thank the authors for their rebuttal. I had a positive opinion of the paper in my initial review, and most of my principal concerns were sufficiently addressed in the rebuttal. After reading the other reviews, I do believe that there is a common interest in having more experiments included. Coupled with my other suggested changes to the current experimental set-up, I think there is still some work to be done in this respect. This also coincides with my wish to see more of your own insights included in the paper, which I think will steer and hopefully also encourage more discussion on this interesting dilemma on where to invest computational resources. Nevertheless, I do not expect such additional experiments to majorly alter the primary ‘storyline’ of the paper, which is why I’m not lowering my current evaluation of the paper. Due to the limited novelty of the technical aspects, I am likewise not inclined to raise my score either, but I think this is a good paper nonetheless. Irrespective of whether this paper is ultimately accepted or not, I definitely hope to see an updated version containing an extended experimental evaluation and discussion.

Reviewer 3



The authors develop a matrix-free version of the BBMM GP inference scheme of [11] to allow scaling to large datasets without running into memory issues. While the ability to scale exact GPs to over one-million training points is impressive, I found that this paper did little to extend the work of [11] since the inference procedure is basically identical except for an MVM approach that computes rows of the kernel matrix on the fly, and a more distributed implementation. That being said, I do appreciate that this implementation could be quite useful in various industrial applications. It would be nice to see reference made to other matrix-free methods considering this is the main innovation of the proposed approach. Nowhere in the paper did it actually describe the method of [11] that you used. The background would be an appropriate place for this. The method used for predictive variance is simply cited in an external reference without any details about the technique in the paper (line 155). Please describe or at least summarize in some detail what the method is within the paper. Additionally, nowhere in the paper is it mentioned that this predictive variance computation is an approximation. This must be explicitly pointed out in the paper considering you are elsewhere claiming you are performing exact GP inference. In particular this must be mentioned wherever you make claims about the speed of the predictive variance computations since it's currently misleading. Lastly, does predictive variance appear in any of the studies? All you results appear to only consider the predictive posterior mean. In the abstract it is claimed that the method uses “more powerful preconditioning” than [11], however, this appears to be incorrect: the preconditioning section clearly states that the proposed approach uses the same partial pivoted Cholesky factorization as was used in [11]. I like the idea of the ablation studies presented, however, I found you claims to be relatively unjustified as a result of the small number inducing points (m) used in the studies. Additionally, it is not clear why you had conducted the sparse GP comparison studies (fig 2) on two small datasets since (all things being equal) it is expected that there is greater redundancy in the larger datasets and so it is expected that greater compression would be possible through the use of sparse GPs on these large problems. I think it would be very interesting to “combine kernel partitioning with inducing-point methods to utilize even larger values of m” as you suggest. I think that your write-off of this idea as a result of figure 2 isn't justified until you consider an m that is much closer in magnitude to n for a large dataset. I think it is reasonable to expect that combining your matrix-free BBMM approach with a sparse GP approximation could be a promising extension to consider. ------------------ (after rebuttal) I thank the authors for their response and for providing the additional test negative log likelihood results. I again commend the authors on the engineering effort required to perform exact GPs on the scale considered. However, in light of very minimal theoretical contributions in the paper, all reviewers appear to agree that this paper's value to the NeurIPS community relies heavily on the quality of the experimental results. After careful consideration, I maintain my position that for this work to be deemed fit for NeurIPS acceptance I would still like to see the empirical evaluations extended to better support the authors' claims. Additionally, I will note that a particular point I was unhappy about in regard to the rebuttal is that the authors never responded to concerns about their unsubstantiated preconditioning claim in the abstract. It appears to me that the authors have simply taken somebody's prior work (i.e. from [11]), modified a free parameter (the rank k), and prominently claimed it as a "more powerful" method in the abstract. This is unacceptably dishonest marketing and I'm surprised there was no commitment to change this in the authors' response.

[Author Response · NeurIPS 2019]

We agree with all three reviewers that evaluating the predictive variances is important. We computed the test negative
log likelihoods (NLL) as R2 suggests which reflect the variances' quality and present the results below. We will include
the NLLs in the final version of the paper in addition to reporting averages and standard deviations in all of our other
tables by running more trials.

| Test NLL | PolTele $(n=9,600)$ | Elevators $(n=10,623)$ | Bike $(n=11,122)$ | Kin40K $(n=25,600)$ | Protein $(n=29,267)$ | KeggD $(n=31,248)$ | CTslice $(n=34,240)$ | KeggU $(n=40,708)$ | 3DRoad $(n=278,319)$ | Song $(n=329,820)$ | Buzz $(n=373,280)$ |
|---|---|---|---|---|---|---|---|---|---|---|---|
| Exact GP | **−0.217** | **0.440** | **0.189** | **−0.456** | **0.897** | −0.895 | **−1.044** | −0.563 | 0.908 | **1.151** | **0.083** |
| SGPR | 0.051 | 0.659 | 0.449 | 0.475 | 0.989 | **−0.986** | 1.419 | **−0.728** | 0.944 | 1.359 | 0.777 |
| SVGP | −0.011 | 0.504 | 0.308 | 0.249 | 1.027 | −0.935 | 1.428 | −0.661 | **0.696** | 1.417 | 0.205 |

**@R1 Regarding fit for NeurIPS:** Thank you for your comments and suggestions. As R2 pointed out, there is a
growing interest in using GPs for large scale problems. As a concrete example, the Bayesian optimization community
within NeurIPS is turning toward large-scale optimization which will greatly benefit from the uncertainty calibration
and flexibility of exact GPs while maintaining prediction times below 1s. GPs have long been a part of NeurIPS and
conventional wisdom has been to compromise performance for scalability by using approximate GPs when $n \gg 1000$.
Our paper shows that we can actually have both when $n < 10^6$ using modern hardware. In our ablations, we also show
the gains from staying non-parametric and keeping inference exact, suggesting future research directions that scale up
exact GP inference. Finally, we will clarify that SGPR is by (Titsias, 2009) and SVGP is by (Hensman et al., 2013).

**@R2 Regarding the implications of this paper:** We agree that we should clarify why we believe these approaches
don't replace sparse GPs. Sparse GPs are still virtually mandatory when (1) $n \gg 10^6$, and (2) for models that require
approximate inference like deep GPs regression or GPs classification. With that said, we agree that this paper raises an
interesting question of when sparse GPs are useful when $n < 10^6$. Realistically, given the enormous hardware demands
of running with $n \sim 10^6$, we expect sparse GPs may still be preferable in that regime. We will move sections of the
paper to the appendix to add this discussion.

Still, we believe that sparse, non-deep GPs could potentially be all but obsolete in the $n \sim 10^4 - 10^5$ regime for
regression, where in fact exact GPs are faster and more accurate than sparse GPs. This has important ramifications, e.g.,
in the large scale Bayesian optimization setting (see @R1). While we agree that many of these improvements are due to
engineering effort, it is hard to overstate the value that engineering effort can provide to the GP community at this point
in time. If this paper demonstrates anything, it is that many of the problems the scalable GP community face may be
solved by sheer engineering effort. For example, we have recently been able to further decrease the running time on a
dataset like 3dRoad ($n \sim 10^5$) from 7 hours to 1 hour through more efficient GPU routines. We believe that this work
will enable researchers and practitioners to use GPs on problems for which it previously would have been intractable.

**@R2 Regarding experimental setup and distributed Cholesky:** Thank you for the suggestions on experimental
setup and the reference on (Nguyen et al., 2019). We are happy to include results with ARD in the supplementary
materials, as well as results on variational versus exact multitask GPs, deep kernel GPs, and other models in the
appendix. Regarding distributed Cholesky, its main drawback are (1) the need to retain a $O(n^2)$ matrix in memory
which may not be feasible and (2) the communication cost. Assuming we actually have $O(n^2)$ memory and have $w$
workers, Algorithm 2 and 5 of Nguyen et al. require two broadcasts of a $b \times b$ matrix block among the workers every
iteration, or $2w \cdot n/b$ exchanges. In contrast, using CG requires exactly $2w$ exchanges to do a linear solve. Furthermore,
a matrix-vector multiplication can be cast as a map-reduce which is simple to implement and embarrassingly parallel.
We were unaware of Nguyen's paper at submission and we will add this discussion to the paper.

**@R3 Regarding predictive variances** We agree that evaluating the variances is important and have done so (see above
table). We note that the precomputation, like CG, can be run to a specified desired tolerance. While in (Pleiss et al.,
2018) the authors were concerned with speed and would only run up to 100 iterations of Lanczos, running hundreds or
even thousands of iterations still affords sufficiently fast and accurate variance computations, making it easy to simply
run Lanczos to within tolerance. We will clarify this.

**@R3 Regarding the number of inducing points:** In (Titsias, 2009) the maximum number of inducing points consid-
ered is 500. Hensman et al. (2013) used 1000 inducing points on the massive Airline dataset. We note that even on
small datasets like kin40k, running with even 4000 inducing points for SVGP would take multiple hours due to the
cubic cost (even with CG) and quadratic number of parameters to fit.

We agree with you that we should improve our explanation for why using larger $m$ is challenging. One reason is that
SVGP requires up to 100x as many linear system solves as exact GPs. For example, training on a dataset with 500k
datapoints and a minibatch size of 1024 results in 48828 solves for 100 epochs. There are other challenges that both
SGPR and SVGP face. In particular, performing solves with the $m \times m$ inducing point kernel matrix $K_{UU}$ requires
either: (1) Cholesky decomposing $K_{UU}$, (2) precomputing $K_{UU}^{-1}K_{UX}$, or (3) solving with $K_{UU}$ in an nested inner CG
loop while solving with $K_{XU}K_{UU}^{-1}K_{UX}$, none of which are easily feasible with our approach due to the associated
space and time complexities when $m$ grows large.

[Meta-Review · NeurIPS 2019]

This paper makes an important contribution to the practice of large-scale Gaussian process inference. Many of the reviewers' concerns were addressed in the rebuttal. However, for the camera-ready version of the paper, I strongly encourage the authors to (1) clarify and include additional implementation details, (2) make clear how their approach differs from past work (if it does differ), and (3) include the promised new experiments with ARD priors and more train/test splits. These changes with help to maximize the paper's impact. Furthermore, please clarify the "more powerful preconditioning" language in the abstract, as suggested by Reviewer 3.